# Pan Aurora Kinase Inhibitor: A Promising Targeted-Therapy in Dedifferentiated Liposarcomas With Differential Efficiency Depending on Sarcoma Molecular Profile

**DOI:** 10.3390/cancers12030583

**Published:** 2020-03-03

**Authors:** Jean Camille Mattei, Corinne Bouvier-Labit, Doriane Barets, Nicolas Macagno, Mathieu Chocry, Frédéric Chibon, Philippe Morando, Richard Alexandre Rochwerger, Florence Duffaud, Sylviane Olschwang, Sébastien Salas, Carine Jiguet-Jiglaire

**Affiliations:** 1Aix-Marseille University, Inserm, MMG, 13005 Marseille, France; Jean-camille.MATTEI@ap-hm.fr (J.C.M.); Corinne.BOUVIER2@ap-hm.fr (C.B.-L.); richardalexandre.rochwerger@ap-hm.fr (R.A.R.); Florence.DUFFAUD@ap-hm.fr (F.D.); solschwang@gmail.com (S.O.); Sebastien.SALAS@ap-hm.fr (S.S.); 2APHM, Hôpital Nord, Service d'Orthopédie et traumatologie, 13015 Marseille, France; 3APHM, Hôpital de la Timone, Service d’Anatomie Pathologique et de Neuropathologie, 13005 Marseille, France; doriane.barets@ap-hm.fr (D.B.); Nicolas.MACAGNO@ap-hm.fr (N.M.); 4Aix-Marseille University, CNRS, INP, Inst Neurophysiopathol, 13005 Marseille, France; mathieu.CHOCRY@univ-amu.fr (M.C.); philippe.morando@univ-amu.fr (P.M.); 5INSERM, UMR 1037, 31077 Toulouse, France; frederic.chibon@inserm.fr; 6APHM, Hôpital de la Timone, Service d’Oncologie adulte, 13005 Marseille, France; 7APHM, Hôpital de la Timone, Département de Génétique Médicale, 13005 Marseille, France; 8Ramsay Générale de Santé, Hôpital Clairval, Institut de Cancérologie, 13005 Marseille, France; 9APHM, Centre de Ressources Biologiques, 13005 Marseille, France

**Keywords:** aurora kinases, liposarcoma, inhibitor, molecular profile

## Abstract

Soft tissue sarcoma (STS) are rare and aggressive tumours. Their classification includes numerous histological subtypes of frequent poor prognosis. Liposarcomas (LPS) are the most frequent type among them, and the aggressiveness and deep localization of dedifferentiated LPS are linked to high levels of recurrence. Current treatments available today lead to five-year overall survival has remained stuck around 60–70% for the past three decades. Here, we highlight a correlation between Aurora kinasa A (AURKA) and AURKB mRNA overexpression and a low metastasis-free survival. AURKA and AURKB expression analysis at genomic and protein level on a 9-STS cell lines panel highlighted STS heterogeneity, especially in LPS subtype. AURKA and AURKB inhibition by RNAi and drug targeting with AMG 900, a pan Aurora Kinase inhibitor, in four LPS cell lines reduces cell survival and clonogenic proliferation, inducing apoptosis and polyploidy. When combined with doxorubicin, the standard treatment in STS, aurora kinases inhibitor can be considered as an enhancer of standard treatment or as an independent drug. Kinome analysis suggested its effect was linked to the inhibition of the MAP-kinase pathway, with differential drug resistance profiles depending on molecular characteristics of the tumor. Aurora Kinase inhibition by AMG 900 could be a promising therapy in STS.

## 1. Introduction

Soft tissue sarcomas (STS) constitute a heterogeneous group of connective tissue malignant tumours involving limbs in 50% of cases, retroperitoneum/trunk in 40% and head/neck soft tissue in 10% [1]. They are rare tumours of a 4–5/10000/year incidence in Europe [2] with high aggressiveness, accounting for 5200 deaths in 2015 in the US [3]. There are over 150 different histological subtypes in the WHO Soft Tissue Tumour classification, with a huge molecular diversity [4]. Molecular aspects have to be depicted to understand biological mechanisms and their heterogeneity (especially their complex genomic profiles) precludes the development of numerous, potentially effective antitumor agents.

Doxorubicin, an intercalating agent blocking nucleic acid synthesis, has been used as the main drug in metastatic STS since the 1980s [5]. Its high haematological toxicity often contraindicates use in elderly population. The five-years overall survival is low, around 60–70% in localized pathology and it is of utmost importance to identify alternative more efficient drugs with less severe adverse effects to improve the prognosis of STS, notably in the elderly.

Mitosis plays a key role in cells proliferation and checkpoints dysregulation may lead to aneuploidy and genetic instability. Aiming for mitosis with generic drugs such taxanes and vinca-alkaloids was validated as the main treatment in several cancers [6,7]. Targeted therapies against molecules involved in mitosis such as cyclin-dependent kinases, polo-like kinases, and aurora kinases were later introduced [8,9,10].

In 2010, Chibon et al. defined the complexity index in sarcomas (CINSARC) signature in sarcomas as a marker of genetic instability [11]. This signature is strongly linked to metastatic evolution. Aurora kinases, part of this CINSARC signature, belong to a highly conserved family of mitotic Serine-Threonine Kinases class. Three members are identified among mammals: Aurora A, B, and C [12,13]. Aurora Kinase A (AURKA) is located at centrosomes, along mitotic spindle microtubules and in cytoplasm. AURKA level is low during G1 and S phases and peaks during the G2/M phase. Phosphorylation at Thr288 in its catalytic domain triggers its kinase activity. AURKA is involved in centrosome separation and maturation, mitosis timing, control and maturation, assembly and stability of mitotic spindle. Its activation and regulation are managed through complex protein phosphorylation, targeted by numerous therapeutics agents [14]. AURKB is located in the middle of the centromere and also peaks during the G2/M phases, while the kinase activity peaks at the transition from metaphase to the end of mitosis. AURKB binds to chromosomes during prophase before relocalizing to the spindle microtubules during anaphase. It plays a key-role in cytokinesis and chromosomal segregation through mitotic spindle fixation control at the microtubule-kinetochore interface. The expression of AURKA and AURKB during the G2/M phase transition is tightly coordinated with histone H3 phosphorylation [15,16] while the overexpression of both kinases is observed in a variety of human cancers [15,17].

These findings led to the development of different drugs like AMG 900, an ATP-competitive small molecule, which inhibits autophosphorylation of AURKA, AURKB, AURKC, and specifically inhibits phosphorylation of Ser10 histone H3, bringing to division blockade and apoptosis, even in antimitotic-drug-resistant tumour cells. AMG 900 is currently evaluated in advanced-stage tumours at adulthood, especially colon, ovarian and breast cancers resistant to conventional chemotherapies [18,19,20].

Recently, Nair and al. assessed the in vitro and in vivo efficiency of an AURK A and B inhibitor, MLN-8237, with a clinical trial confirmation [19,21]. These findings comforted the use of a pan AURK inhibitor in our study. However, the impact of a pan AURK inhibitor in correlation with expression of AURKA and B was never studied in liposarcomas (LPS), the most frequent STS type. Mechanism of action of this drug is not well understood. We report in this work the effect of AURKA and AURKB inhibition by AMG 900, doxorubicin or both in different sarcoma cell lines, focusing on 4 dedifferentiated liposarcomas. The mode of action of this inhibitor was also considered through kinome analysis, specifying the kinase inhibition profile depending on tumour drug resistance characteristics.

## 2. Results

### 2.1. Survival Curves Specific to AURKA and AURKB mRNA Expression

The expression of Aurora kinase A and Aurora kinase B mRNA was studied in a cohort of 339 patients from the FSG database [11]. Patients were selected for the presence of a tumour exhibiting a complex genomic profile, meaning without any recurrent chromosomal translocation (Figure 1A,B). Tumours were sorted according to the expression level of *AURKA* or *AURKB* mRNA. A worse prognosis (metastasis-free survival) was linked to an overexpression of *AURKA* and *AURKB* mRNAs, with a *p log rank* of 3.31.10^-6^ and 0.0224 respectively (Data available at FSG: http://atg-sarc.sarcomabcb.org/atg_sarc.php). TCGA-SARC project from the TCGA database was analysed for AURKA and AURKB mRNA expressions (Appendix A). The TCGA-SARC propose a cohort of 206 patient with heterogeneous STSs. Interestingly, overexpression of AURKB mRNA correlates with lower overall survival (OS) (logrank *p* = 0.0036). Regarding disease free survival (DFS), the overexpression of AURKA and AURKB mRNAs is associated with recurrence (logrank *p* = 0.021 and 0.00064 respectively) (http://www.cancer.gov/about-nci/organization/ccg/research/structural-genomics/tcga).

### 2.2. Aurora Kinase Expression in Soft Tissue Sarcomas Cell Lines

#### 2.2.1. CGH and Gene Expression

Nine cell lines derived from sarcomas were assessed through comparative genomic hybridization. Results are presented in Figure 1C with corresponding histological subtype and tumour location. A gain at *AURKA* locus for IB105, IB111, IB112, IB116, IB119, LPS78 and LPS80 was detected. No variation was observed for IB115 and MFH152. *AURKB* amplification was absent in IB105, IB115, IB116, IB119 and LPS80. A gain was observed in IB112, and a loss at AURKB locus was observed in IB111, MFH152 and LPS78. mRNA expression was analysed in the nine cell lines and compared to a standard skeletal muscle mRNA (Figure 2A). Tumour cell lines exhibited higher expression level of Aurora kinase mRNAs by more than 300-fold in IB105, IB112, IB116, and LPS80 compared to commercial standard skeletal muscle. IB115, IB119, LPS78 and MFH152 *AURKA* expression ranged from 170 to 250-fold the standard. Only IB111 presented a moderate *AURKA* expression of 2.5 more than the standard RNA. *AURKB* expression was more than 5,000 times higher in all tumour cell lines except IB111 where the increase was of 200. RNA was then purified from safe skeletal muscle biopsies of 10 patients to evaluate the level of *AURKA* and *AURKB* RNA in normal tissue. It appeared that *AURKA* and *AURKB* RNAs were expressed at the same levels compared to standard RNA (Figure 2B).

#### 2.2.2. Protein Expression

After protein extraction and quantification, Aurora kinase A and B protein levels were measured in STS cell lines (Figure 2C). AURKA was expressed at a high level in IB105, IB119 cell lines, intermediate in IB115, IB116, and MFH152 and low in IB111, IB112, LPS78, and LPS80. High levels of AURKB were detected in LPS78, LPS80, IB105, IB112, IB119, and MFH152.

### 2.3. Impact of Inhibition of Aurora Kinases A and B RNAs on Liposarcoma Cell Survival

The role of AURK A and AURK B was studied in dedifferentiated LPS (DDLPS) by performing interference RNA with siRNA on IB111, IB115, LPS78 and LPS80 (Figure 3A). The efficiency of two different siRNA sequences was first validated for downregulating each targeted gene (Figure 3A). The cell survival was analyzed by MTT assay 72h after transfection (Figure 3B). *AURKA* and *AURKB* targeting with siRNAs led to an inhibition of their mRNA target in all the cases. The inhibition of Aurora kinases RNAs caused a significant mortality of 63% (siA1), 58% (siA2), 51% (siB1) and 55% (siB2) in IB111. In IB115 and LPS78 cell mortality ranged from 47% to 59%. No significant effect was observed in LPS80.

### 2.4. Characterisation of AMG 900 Impact on LPS

#### 2.4.1. Cytotocic Effect of AMG 900 and Doxorubicin on Sarcoma Cell Lines

To decipher AURKA and AURKB roles in LPS tumorigenesis, both were inhibited by a pan AURK inhibitor: AMG 900 (Figure 4A). Several concentrations were tested to determine the inhibitory concentration leading to 50% of mortality (IC50). Same was performed with doxorubicin (Figure 4B). IB111 cells appeared as more resistant to AMG 900 compared to IB115, whereas IB115 was more resistant to doxorubicin. LPS78 and LPS80 showed an intermediate response to AMG 900. LPS80 appeared as the most sensitive to doxorubicin (Figure 4C).

#### 2.4.2. Tumorigenicity

KI67 marker was used to measure the proliferative index of LPS78 and IB115 when treated or not with AMG 900. No significant effect was observed in the two cells lines. The impact of AURKA and AURKB on migration process and clonogenicity was then analysed after AMG 900 treatment. No significant effect on migration was observed either for LPS78 or IB115 with transwell migration assay. Clonogenicity was impaired in the two cell lines for all concentrations tested (Figure 5A). For LPS78, AMG 900 reduced clonogenicity from 10% of the IC_50_, and a higher concentration did not lead to a higher reduction. IB115 clonogenicity was less impacted, as a dose response and low percentage of the AMG 900 IC50 did not lead to an inhibition of clonogenicity.

#### 2.4.3. DNA Fragmentation is Induced by AMG 900, Doxorubicin and AURKs siRNA

To analyse the apoptotic and necrotic effects of AMG 900 on LPS, DNA fragmentation was quantified after 72h of treatment. IB111 apoptosis in AMG 900 condition was increased by 2.37-fold versus control and 4-fold with doxorubicin treatment. IB115 presented the same level of apoptosis with AMG 900 or doxorubicin treatment, 3.72 and 4.21 more than control condition, respectively. LPS78 showed the strongest apoptosis level of 4.26-fold with AMG 900 and 2.36-fold with doxorubicin (Figure 5C). To confirm the implication of Aurora kinases in this process, RNA interference was performed in LPS78 before DNA fragmentation analysis. Each siRNA induced a significant apoptosis between 2.41 and 2.82 more than control condition. Apoptosis was weaker with RNAi experiment compared to AMG 900. The percentage of apoptotic and necrotic cells was determined with annexin V and propidium iodide staining on IB115 cells. When treated 72h with AMG 900, early apoptosis is 1.8-fold stronger than in control and late apoptosis/necrosis is 5.66 stronger compare to control (Appendix A).

#### 2.4.4. Cell Cycle is Differentially Modulated by AMG 900 in Different Liposarcoma Cell Lines

AURKA and AURKB have their main functions into cell cycle. To determine AURK inhibition influence on each phase of cell cycle, cells treated with siRNA, AMG 900 or doxorubicin were analysed after propidium iodide (PI) labelling (Figure 6A). In IB111 and LPS78 60.7% and 56% of cells were in G0/G1 phase respectively. *AURKA* or *AURKB* RNA inhibition led to a decrease about 2.3 and 3.3 times. The doxorubicin and AMG 900 treatments induced a decrease about 3.7 and 20.3 time compared to the control condition. In IB115, the G0/G1 phase was less represented with 26% of the population. The same effect was observed using RNAi or drug treatment, the doxorubicin induced the strongest diminution of 4 times the control condition. In all cell lines, S phase was impaired with RNAi or AMG 900 (strongest effect). Doxorubicin had no effect on IB115, but increased the percentage of cells in S phase for IB111 or LPS78. G2/M phase was increased for IB111 with all treatments, RNAi, AMG 900 and doxorubicin. IB115, where G2/M represents the major phase (39.6%), treatments did not induce notable variations, except AMG 900 (reduction to 22.3%). LPS78 showed an increased cells percentage when treated with RNAi or doxorubicin as well as a severe decrease with AMG 900 treatment. Basal level of polyploidy was variable in LPS cell lines: IB111 and LPS78 had low proportions of polyploid cells, 5.9% and 6.2% respectively, versus 23.2% for IB115. RNAi treatment increased the number of polyploid cells in all cases, with a strongest impact of *AURKB* RNAi versus *AURKA* RNAi. Doxorubicin also induced polyploidy and the strongest effect was observed in IB115. AMG 900 led to the most important variations of this phase. LPS78 polyploidy was increased by 11.2-fold, 5.9 in IB111 and 2.7 in IB115. LPS78 AMG 900-induced polyploidy is correlated to an increase of nucleus size (Figure 6B).

#### 2.4.5. Kinomic Analysis: AMG 900 Action is Linked to MAPK Kinase Modulation

In order to understand the molecular regulation induced by AMG 900 treatment, IB111 and IB115, two cell-lines with significantly different sensitivities to AMG 900 treatment, were analyzed for pan kinase activity (Figure 7A). IB115 is described as a sensitive cell line regarding AMG 900 IC_50_, whereas IB111 is resistant. We did not observe significant variation of the Phospho-Tyrosine Kinase (PTK) signal between sensitive and resistant cell lines when treated for one hour with AMG 900 or not. A strong inhibition of the serine threonine kinase (STK) activity was pointed out on the MAP kinase P38, ERK1, ERK2 and Jun-Kinase (JNK) in IB115 when treated at the AMG 900 IC_50_. Inhibition of several members of the CDK family was observed, without activation under treatment. IB111 cell line showed both inhibition and activation of different pathways. AKT1, AKT2, AKT3, pKC, CAMK and CDK1, CDK2 were activated in response to AMG 900, while some CDK members and two MAPK family members were inhibited.

We focused on the MAP Kinase profile variations between sensitive and resistant cell lines that was analysed by Western blot phosphorylation of MAP Kinases ERK1/2, P38. and JNK. In sensitive cell line IB115, phosphorylation of MAPK was significantly upregulated for ERK1, ERK2, and p38 kinases, while IB111 did not show significant regulation of these proteins (Figure 7B).

### 2.5. Combination of AMG 900 and Doxorubicin Treatments

AMG 900 was tested in combination with doxorubicin (Figure 8). Survival was tested in cell treated with AMG 900 or with a combination of IC_50_ doxorubicin (cell dependant) plus a range of AMG 900 from 1.10 × 10^−3^ to 1.10 × 10^−7^ M (mol/L). IB111 and LPS78 showed a combined effect in the presence of both drugs, with a better cytotoxic effect on IB111. The addition of doxorubicin did not enhance toxicity on cells in LPS80 a minimum dose of 1µM was required to induce a visible effect in combination with doxorubicin. IB115 presented an antagonist profile as each drug is efficient alone but a combination of AMG 900 and doxorubicin reverses the cytotoxic effect on cells.

## 3. Discussion

Soft tissue sarcomas are tumours of poor prognosis due to the weak efficiency of standard chemotherapy. An important reason might be the histological heterogeneity into STSs with numerous types and subtypes. We focused the study on dedifferentiated liposarcomas belonging to the most frequent sarcomas subtype also gathering well-differentiated liposarcomas. Those tumours present a high risk of recurrence due to incomplete resection in retroperitoneal tumour and high metastases dissemination risk. Dedifferentiated LPS are more aggressive than well differentiated LPS. Both present MDM2 and CDK4 amplification but DDLPS also accumulate nonspecific genomic aberrations. Some studies explored the interest of MDM2 [22] or CDK4 [23] inhibitors but none showed strong therapeutic effects in LPS. CINSARC signature offers new targets to explore [11], revealing Aurora kinases A and B genes were significantly linked to poor prognosis (Figure 1A–B). Recently, a dual inhibition of AURKA and AURKB shown efficiency on several cancer cell lines, including sarcomas [21] and a promising Phase II study was done on metastatic sarcomas [19].

Here, we observed a different correlation between overall survival, progression-free survival, metastatic recurrence and AURKA and AURKB mRNA expression (Figure 1A; Appendix A). AURKA is highly correlated to metastatic recurrence maybe due to its capacity to induce MMP2 expression via AKT/NF-KB [24] and its role in epithelial–mesenchymal transition [25]. Whereas AURKB is indifferently related to OS, PFS and metastatic recurrence maybe due to its capacity to induce polyploidy and DNA damage accumulation [26]. We then confirmed the heterogeneous expression of AURKA and AURKB in nine sarcoma cell lines, both at genomic and protein levels (Figure 2A-C). The expression of Aurora kinase A and B mRNAs was low in normal muscle tissue compared to sarcoma cell lines (Figure 2B). Among the same histological subtype, dedifferentiated LPS presented variations. STS cell lines exhibited a heterogeneous range of response to AMG 900 or doxorubicin IC_50_ (Figure 4). We tried to link these data with expression rates, however no correlation was found at either mRNA or protein levels. The same observation was reported in breast cancer, where AMG 900 efficacy was not linked to AURKA protein and mRNA expressions or classic molecular specific subtype [27]. Another study focusing on Alisertib in colorectal cancer cell-lines, an AURKA inhibitor, also evoked a variability in cellular response independent of the genetic status of main known prognosis genes [20].

The inhibition of AURKA and AURKB with siRNA impairs the cell survival whatever the level of gene extinction and induces apoptosis (Figure 3B). Drug targeting by AMG 900 leads to same effects and we noticed a weaker apoptosis in RNAi assay, possibly due to the inactivation of a single target, compared to AMG 900, targeting all the AURKs.

We did not observe significant modification of the proliferation index, Ki67, with AMG 900 while clonogenic proliferation was decreased even with low AMG 900 concentration (Figure 5A). In cell-cycle, LPS78 and IB115 shown less than 10% cells in G0/SubG1 phase, suggesting that Ki67 is expressed in more than 90% of cells in other cell cycle phases without any treatment [28] (Figure 6A).

Previous studies pointed the non-mitotic role of AURKA in cell migration. Inhibition of AURKA by RNA interference or alisertib in normal fibroblast-like [29] or epithelial ovarian cancer [30] cells induces a reduction of the migration process. In the present model no variation in cell migration was observed (Figure 5B). In cell cycle analysis, IB111 and LPS78 cell lines showed an arrest in G2/M as known [31,32]. On the contrary, IB115 cells accumulated in G2/M phase and mostly the polyploid phase (Figure 6A). This could be consistent with the low AURKB protein expression in IB115 as AURKB mutation or inhibition is responsible for polyploidy [26]. Furthermore, at the basal stage, IB115 cell line exhibited numerous polyploid cells.

AMG 900 treatment or RNA interference led to the accumulation of IB115 and LPS78 in G2/M and polyploid phases. A highest rate in polyploid phase was visible for AMG 900, suggesting that the pan-aurora inhibition is more deleterious for cells than the siRNA single target. This result is linked to an increase of the nucleus size in LPS78 (Figure 6B). For IB111, cells accumulate in the G2/M phase and are less polyploid. It might reflect the resistance of IB111 to AMG 900 treatment, being stopped before mitosis preventing mutations able to induce apoptosis. In those tumour cell lines, a variability was also noticed in an additional phase of polyploid cells.

Regarding response to AMG 900, kinomic analysis highlighted distinct profiles in IB115 and IB111 (Figure 7A). The sensitive cell line IB115 harboured modifications of the different MAPK protein activities whereas IB111 very few. Western blot analysis confirmed a difference in MAPK phosphorylation between the two cell lines (Figure 7B). Interestingly, AMG 900 treatment is linked to ERK1/2 and p38 up-phosphorylation in sensitive IB115, probably related to an inactivation of these kinase and a sensitivity to drug treatment. MAP Kinase activation is a very tightly regulated process, the sub-cellular localization being driven by interaction with scaffolding proteins and the duration of activation being regulated by molecular interaction with partner proteins [33]. ERK1/2 are mostly known to induce survival signal, and under specific conditions, ERK phosphorylation is able induce apoptosis [34,35], whereas Phospho-p38 is known to be mainly involved in apoptosis (for review, see [36]). If not translocated to the nucleus, P-ERK mediates cytosolic signals. PED/PEA-15 is one of the major cytosolic regulators of ERK activity and blocks ERK dependent transcription but not its cytosolic counterpart [37,38]. Interacting together, these proteins PED/PEA-15 and ERK mutually inhibit their interaction with other partners. Both ERK and Phospho-ERK are relocated into the cytoplasm and to do not induce proliferative and/or survival signals. PED/PEA-15 could not interact with DISC allowing apoptotic signalling via caspase3 [39,40]. All the MAPK pathways are blocked by AMG 900 treatment probably stabilizing interactions between MAPK and scaffolding proteins. In the resistant cell line IB111, kinomic analysis showed activation of several partners of the anti-apoptotic signalling mediated by PED/PEA-15: AKT members, pKC and CAMs are involved in PED/PEA-15 activation leading to DISC inhibition [41,42]. We could suppose here that PED/PEA-15 is a key regulator for AMG 900 resistance in LPS.

Global dose effect confirmed the awaited efficiency of AMG 900 and is encouraging for conducting animal experiments to confirm the results. However, these data also revealed heterogeneity regarding in vitro drug response. In some cell-lines, the high response at low doses, compared to doxorubicin, is a good indicator to test AMG 900 as monotherapy. Moreover, similar efficiency was observed in other STS cell-lines when combined with doxorubicin. This should help to decrease doxorubicin doses thus reducing adverse effects, especially in the elderly, or to open new lines of treatment in refractory tumours. These findings may lead to investigate the criteria linked to the response rate, which could help to decide the best regimen for patients. Molecular characterization of each tumour could become mandatory before treatment, because the specific genetic heterogeneity of sarcomas appears likely linked to drug efficiency.

No gain was observed when increasing AMG 900 concentrations combined to IC_50_ dose of doxorubicin. These findings suggest a possible competition between AMG 900 and doxorubicin in some tumours because of their genetic alterations resulting in drug bypasses/resistance. In addition, primary or secondary substrates of AURKA/B might be altered by doxorubicin. Moreover, it appears that doxorubicin DNA alters apoptosis is ERK dependant. ERK inhibition leads to enhanced drug resistance [43,44]. In sensitive IB115, the combination AMG 900/doxorubicin appeared less efficient than each drug alone. The inhibition of ERK activity by AMG 900 could explain this antagonist effect, whereas no similar effect was observed in the resistant and ERK active IB111.

Although these mechanisms were never explored in sarcoma, other cancer models showed different impacts of aurora kinases inhibitors depending on the expression level of certain genes such *MYC* in lung cancer [45,46]. This classic variability in drug response led some authors to promote cell profiling in order to evaluate potential efficiency of drugs [47].

Considering AURKA and AURKB inhibitors as promising anticancer drugs, possible predictive response factors as p53 (suggested to induce a post-mitotic arrest when AURKB is inhibited) are to be further explored [48,49] The effect was clearly established in breast cancer cell-lines with high efficiency of AMG 900 in p53-loss-phenotype cells [27]. In our study, p53 status was not linked to a specific response pattern. Furthermore, AMG 900 remains a pan aurora kinase inhibitor, which may impact several secondary substrates.

## 4. Methods

### 4.1. Subject and Samples

Subject data and samples were obtained from FSG database (French Sarcoma Group database) (http://atg-sarc.sarcomabcb.org/), which is part of the Conticabase (www.conticabase.org), containing data from adult soft tissue sarcomas treated in 11 centers. Deltoid biopsies of patients were provided by the AP-HM tumour bank (authorization number: AC-2018-3105).

### 4.2. DNA Extraction, Array Comparative Genomic Hybridization Analysis and TCGA Database

Briefly, genomic DNA was isolated with a standard phenol-chloroform extraction protocol. Array-CGH experiments were performed with DNA microarray developed in Institute Bergonié laboratory as previously described [50] and data were analyzed with a software developed at Institut Curie (CAPweb, http://bioinfo-out.curie.fr/CAPweb/) [51].

Analysis of RNA expression from the TCGA database was done by using GEPIA ([52], http://gepia.cancer-pku.cn/index.html). We analysed *AURKA* and *AURKB* expression prognostic value (OS and PFS) for the TCGA-SARC cohort.

### 4.3. STS cell Lines Establishment and Culture

STS cell-lines were developed at Institut Bergonié, France. Each cell line was derived from a tumour patient bearing soft tissue sarcoma. Sample tumours were dissociated by mechanistic and enzymatic action. Cell lines genomic stability was done by CGH each 5 passage over at least 20 passages. For experiments cells were grown in RPMI (PAA Laboratories, Les Mureaux, France) supplemented with 10% Fetal Bovin Serum (Gibco, Saint Aubin, France), 50U/mL penicillin, 50 mg/mL streptomycin (Gibco), 1mM L-glutamine and 1mM sodium pyruvate (Gibco) and maintained at 37 °C with 5% CO_2_.

### 4.4. MTT Assay

One day before treatment, 20.000 cells were seeded in 24-well plate. The next day drug was added at specified concentrations or siRNA transfection was done. 72 h after treatment, MTT (3-(4,5-dimethylthiazol-2-yl)-2,5-diphenyltetrazolium bromide) (Sigma, Saint-Quentin Fallavier, France) was performed following manufacturer recommendations. Doxorubicin and AMG 900 were obtained for Sigma and Euromedex (Souffelweyersheim, France), respectively. siRNA transfection was done with Lipofectamine (ThemoFischer Scientific, Illkirch-Graffenstaden, France) according to manufactured protocol. The Silencer^®^ Select validated siRNA sequences were purchased from Ambion: s196, s197, s17611, s17612 and negative control n°1). Cells were harvested 48h after transfection for q-RT PCR analysis.

### 4.5. q-RT PCR

RNA was first extracted using guidelines and Qiagen^®^ extraction kit (Ref52304) and dosed to allow first step of denaturation with 1μg total RNA, 0.2 μg/μL of Random hexameres (ThemoFischer Scientific), 5 mmol/L of DNTP (Sigma) and DNAse RNAse free H_2_0 (Gibco) −70 °C, 10min. Reverse Transcription (RT) was performed with SuperScript RT II (Invitrogen) according to manufacturer’s instructions, at 42 °C for 2 h. Two μL of RT solution were used for the quantitative PCR. LightCycler ^®^ 480 SYBR Green I Master (Roche) was used for PCR. q-PCR primers sequence used are: AURKA FW 5′TTCATAGAGACATTAAGCCAGAGAAC3′; AURKA RV 5′CCACAGAGAGTGGTCCTCCT3′; AURKB FW 5′GATGGCCCAGAAGGAGAACT3′; AURKB RV 5′AGGCTCTTTCCGGAGGACT3′; Primer sequences of reference genes: GAPDH FW 5′CAAATTCCATGGCACCGTC3′; GAPDH RV 5′CCCACTTGATTTTGGAGGGA3′; 18S FW 5′CTACCACATCCAAGGAAGGCA3′, 18S RV 5′TTTTTCGTCAACTACCTCCCCG3′. PCR program for 18S is: Denaturation step [95 °C-10 min], PCR step [Denaturation 95 °C-15 sec- Hybridization 67 °C-30 sec] ×35 cycles followed by fusion curve step [Denaturation 95 °C-5 sec- Hybridization 70 °C-1 min]. PCR program for the other genes is: Denaturation step [95 °C-10 min], PCR step [Denaturation 95 °C-15 sec- Hybridization 65 °C-20 sec] ×45 cycles followed by Fusion curve step [Denaturation 95 °C-5 sec- Hybridization 70 °C-1 min]. Standard skeletal muscle RNA was purchased from Clinisciences (Biochain, Nanterre, France).

### 4.6. Western Blot

Cells were expanded and then collected after 10 min of trypsin-EDTA digestion at 37 °C. About 5 × 10^6^ PBS-washed cells were lysed in 50 μL of protein extracting buffer (50mM Tris-HCl pH 7.4, 250mM NaCl, 5 mM EDTA, 1 mM DTT, 1% Triton X100, 0.1% SDS, 0.5% DOC and 1% NP40) with ultrasound dissociation then incubation 20min-on-ice. Cell fragments were cleared by centrifugation (10,000 rpm, 10 mins, 4 °C). Proteins amount was determined by BCA assay (Sigma). Ten μg of denaturated proteins were loaded on a 12% SDS-PAGE gel. Western blot was blocked with 5% BSA-TBS 0.1% Tween 20 for one hour. Primary antibodies were incubated overnight at 4 °C Fluorescent secondary antibodies (donkey anti goat-800CW or donkey anti mouse-680LT from LI-COR) were detected with Odyssey^®^ CLx Imaging System from LI-COR. Goat anti-human AURKA antibody (1/1000, SAB2500135), mouse anti-human AURKB antibody (1/500, WH0009212M3) were purchased from Sigma, Phospho MAPK kinase (1/2000 ERK1/2; 1/1000 p38 and SAPK/JNK) were purchased from Cell Signaling Technology and mouse anti-human GAPDH antibody (1/1000, 5G4) obtained from HyTest Ltd. Significant differences between values obtained in control groups and different treatment groups were determined by the Mann–Whitney test. *p* < 0.05 was assigned significance.

### 4.7. Clonogenic Assay

One day before treatment, 1000 cells were seeded in 10cm petri dish. The following day, IC_50_ AMG 900 was added for 24 h. Eight days after treatment, cell colonies were stained with crystal violet and counted with ImageJ software.

### 4.8. Migration Assay

100.000 cells, in serum free medium +/− AMG 900, were plated in transwell. Transwells were deeped in 24-well plate containing 10% serum medium. Cells were allowed to migrate for 24 h then fixed in paraformaldehyde (4%) before being labelled by 0.1% crystal violet. Labelled cells were solubilized in SDS 10% and DO was quantified by spectrometry at 570 nm [53].

### 4.9. Proliferation Assay

One day before treatment, 50.000 cells were seeded in Lab-Tek^®^ Glass Chamber Slide. The next day IC_50_ AMG900 was added for 72 h. Cells were then fixed with paraformaldehyde (4%) and immune-labelled with Ki67 antibody (MIB1-1/100, Dako, Trappes, France). Secondary FITC-goat anti-mouse (1/100, 115-095-146, Jackson Immunoresearch, Cambridgeshire, UK) was used to performed imaging by fluorescent microscopy (AxioObserver, Carl Zeiss, Marly le Roi, France).

### 4.10. Cytometry Analysis

Cells were prepared as for MTT assay regarding DNA fragmentation and cell cycle analysis. Instead of MTT procedure, cells were harvested after 10 min of trypsine-EDTA digestion at 37 °C and were fixed and permeabilized with 70% ethanol. After PBS wash, cells were labelled with PBS supplemented with RNAse A (50 μg/mL) and propidium iodide (40 μg/mL) as previously described [54]. Cells were incubated in the dark for 30 min and rapidly analysed with a FACS Calibur cytometer (BD Biosciences, Le Pont de Claix, France). Results were computed with the CellQuest Pro Software (BD Biosciences). Propidium iodide/Annexin V assay was performed with the FITC Annexin V Apoptosis Detection Kit for BD Biosciences as recommended by the manufacturer.

### 4.11. Kinome Assay

Serine/Threonine and Tyrosine kinase activity was challenged with AMG 900 and analysed on PamChip^®^ with the Pamstation^®^12 (PamGene, BJ's-Hertogenbosch, The Netherlands). PamChip^®^4 contains 4 arrays with 144 target peptides that can be phosphorylated by kinases [55,56]. Cells were treated for one hour or not with AMG 900, then extracted in M-PER buffer (ThemoFischer Scientific) containing inhibitor cocktails (Halt Phosphatase Inhibitor Cocktail and Halt Protease Inhibitor Cocktail EDTA free 1/100, Pierce). Ten µg of protein was loaded on 4 arrays of the PamChip^®^. Phosphorylation activity was tagged by a FITC-conjugated antibody and recorded with a CCD camera and the Evolve software v. 1.2 (PamGene). Results were analysed with BioNavigator (Pamgene). Experiments were performed in triplicate.

## 5. Conclusions

Our analysis shows that there is a link between metastatic-free survival and the overexpression of AURKA and AURKB messenger RNA. Our findings highlight cytotoxic effect of AMG 900 on several STSs, especially on dedifferentiated LPS, with an impact on clonogenicity, proliferation and DNA accumulation, acknowledging that Aurora Kinase inhibition by AMG 900 could be a promising therapy in STSs. Drug mode of action seems to be linked to MAP-kinase pathway on kinome analysis with differential response depending on sarcoma molecular profile.

## Figures and Tables

**Figure 1 cancers-12-00583-f001:**
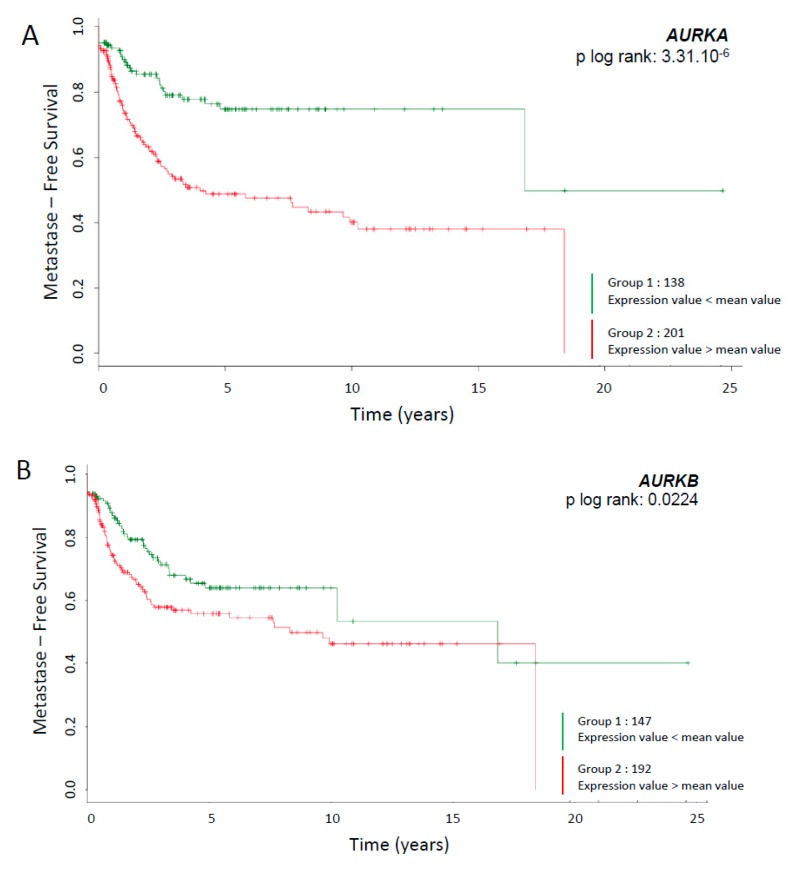
Metastasis-free survival analysis according to mRNAs AURKA and AURKB expression and characteristics of STS cell lines. (**A**) Metastasis-free survival analysis according to mRNA AURKA expression. Subjects are stratified in two groups with significantly different metastasis-free survival rate (MFS rate; *y* axis) during the time after diagnosis (*x* axis). Subjects with the lowest expression compare to mean expression are in red, and those with the highest one are in green. (**B**) Metastasis-free survival analysis according to mRNA AURKB expression. P values correspond to the log-rank test comparing the survival curves. (**C**) Histological type, grade, localization and CGH statute for AURKA and AURKB are listed in the table. G: Gain (2–10 copies of gene present); L: loss (1 copy of gene present); N: no variation (2 copies of gene present). NC: not communicated.

**Figure 2 cancers-12-00583-f002:**
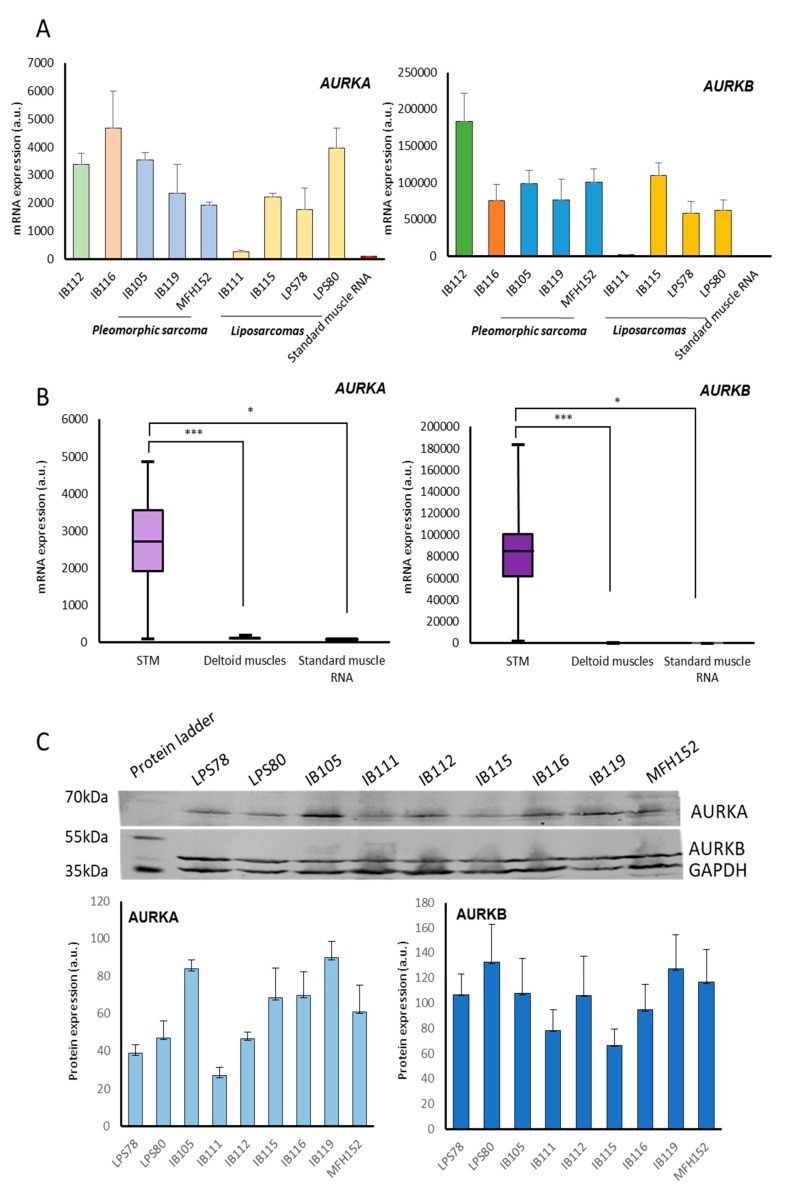
Characterisation of AURKA and AURK B expressions in STS cell lines and normal tissues. (**A**,**B**) Gene expression of *AURKA* and *AURKB*. After RNA extraction and reverse transcription, mRNA expressions are analysed by q-RT PCR. Expressions are normalized to the standard RNA. (**A**) Gene expression of *AURKA* and *AURKB* are compared in STM and a standard RNA. STS cell lines overexpressing mRNAs of *Aurora kinases A* (left) and *B* (right). (**B**) Gene expression of *AURKA* and *AURKB* are compared between STS cell lines, normal tissue from 10 deltoid biopsies and a standard RNA. Tumour cell lines still overexpressing mRNA of *AURKA* and *AURKB*. Histograms represent mean values of triplicate +/− SEM. (*: *p* < 0.05; ***: *p* < 0.001). (**C**) Protein expression of AURKA and AURKB in STS. After protein extraction, AURKA and AURKB protein expressions were quantified by Western blot. The blot is representative of three independent experiments. Histograms represent mean of protein expression values normalized by GAPDH protein expression +/− SEM. More details of western blot, please view at the Appendix A.

**Figure 3 cancers-12-00583-f003:**
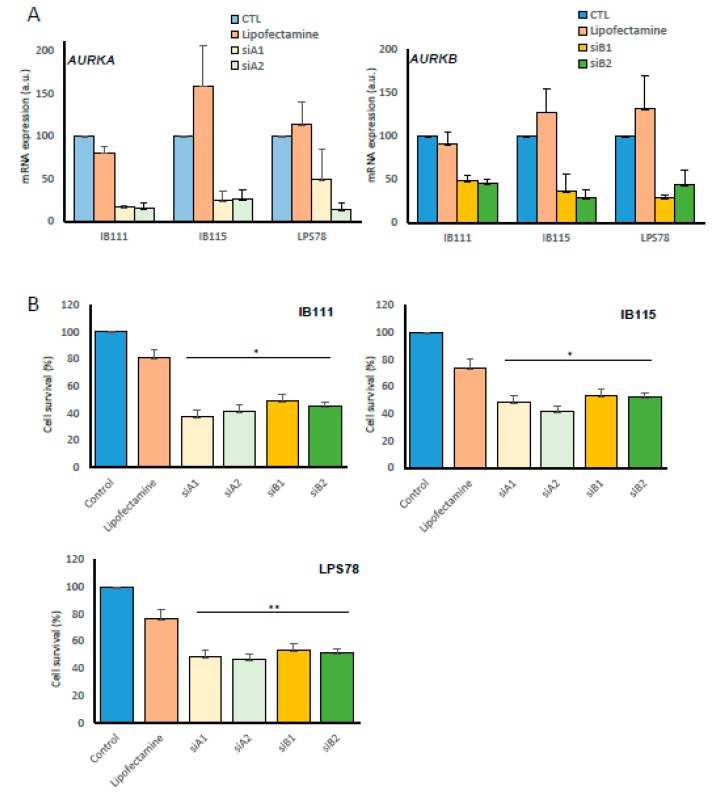
Aurora kinases A and B interfere with LPS cell survival. Three LPS cell lines were challenged with siRNA targeting aurora kinases A and B mRNA. (**A**) Efficiency of RNA silencing. Cells are lipofectamine-transfected with two different siRNAs by gene. After 48h targeted mRNA level of *AURKA* and *AURKB* mRNA was determined by q-RT PCR. (**B**) 72h after transfection, cell survival was assayed with MTT assay in the LPS cell lines. Histograms represent mean values of independent experiments done in duplicate or triplicate +/− SEM. (*: *p* < 0.05; **: *p* < 0.01).

**Figure 4 cancers-12-00583-f004:**
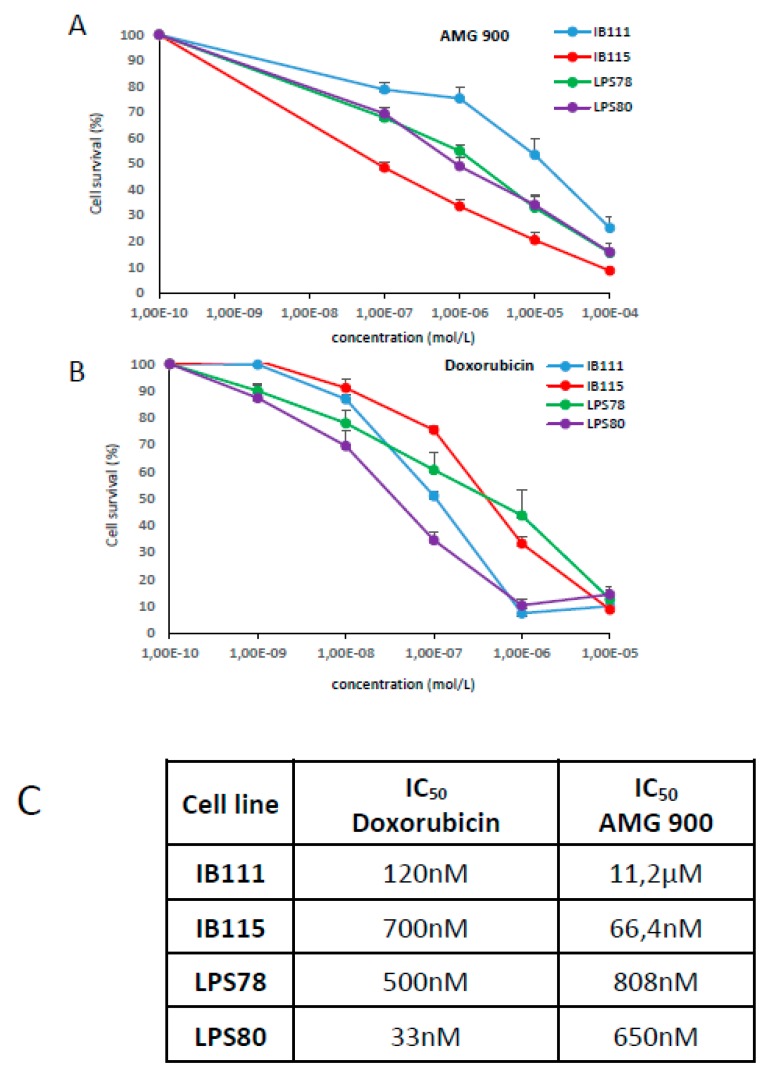
Comparative cytotoxic effects of AMG 900, a pan AURK inhibitor and the doxorubicin on LPS. Four LPS cell lines were challenged for 72 h with increasing doses of drug (**A**: AMG 900, **B**: Doxorubicin) to determine the corresponding IC_50_ on cell survival. After treatment, surviving cell were quantified with MTT assay. Graphs represent mean values of at least three independent experiments +/− SEM. (**C**) Recapitulative table of IC50 for LPS cells lines.

**Figure 5 cancers-12-00583-f005:**
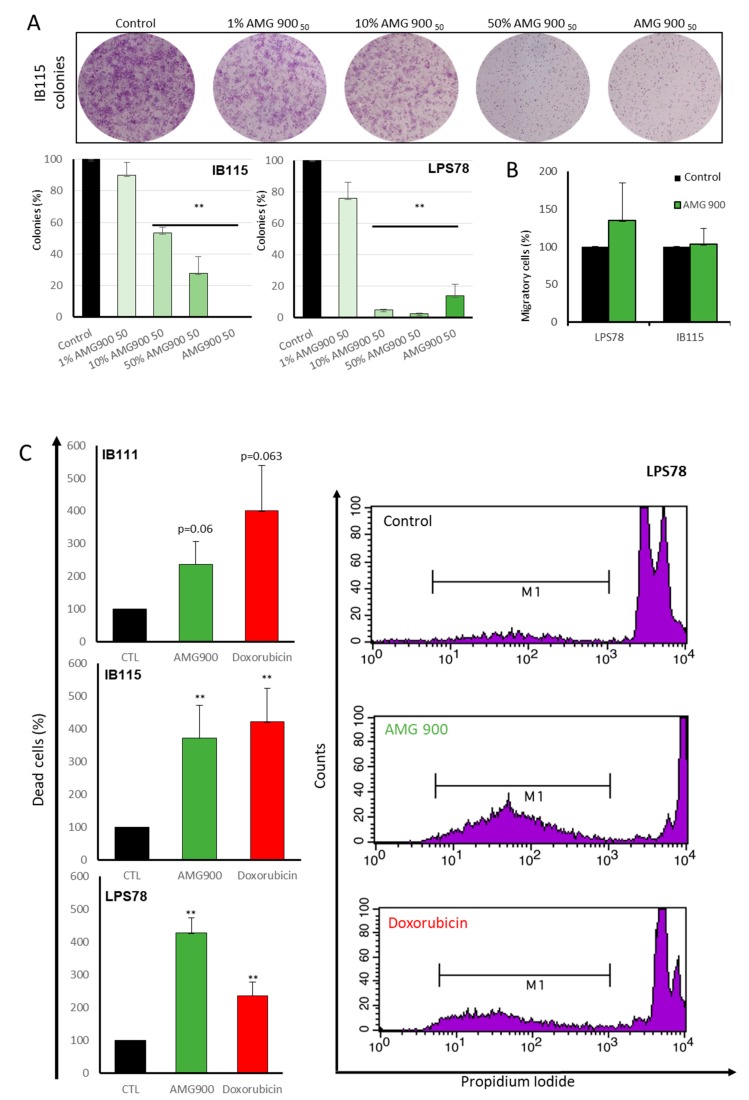
AMG 900 impairs clonogenicity and induces apoptosis differently in LPS. **(A**) Clonogenicity in two different LPS cell lines. Cell are plated at low density and treated for 24H with different percentage of AMG 900 IC_50_ and clones are counted after crystal violet coloration at 8 days of treatment. Histograms are mean values of triplicate +/− SEM. (**B**) Migration in LPS78 and IB115 cells. Cells were plated with serum free medium in a transwell and dive into serum containing medium (+/− AMG900) for 24H before crystal violet coloration and counting. (**C**) Apoptosis and necrosis induce by AMG 900 or doxorubicin in three different LPS cell lines. Cells are treated for 72H before analysis. Apoptosis was determined by flow cytometry quantification of cell lines PI labelled. Histograms are mean values of triplicate +/− SEM (**: *p* < 0.01).

**Figure 6 cancers-12-00583-f006:**
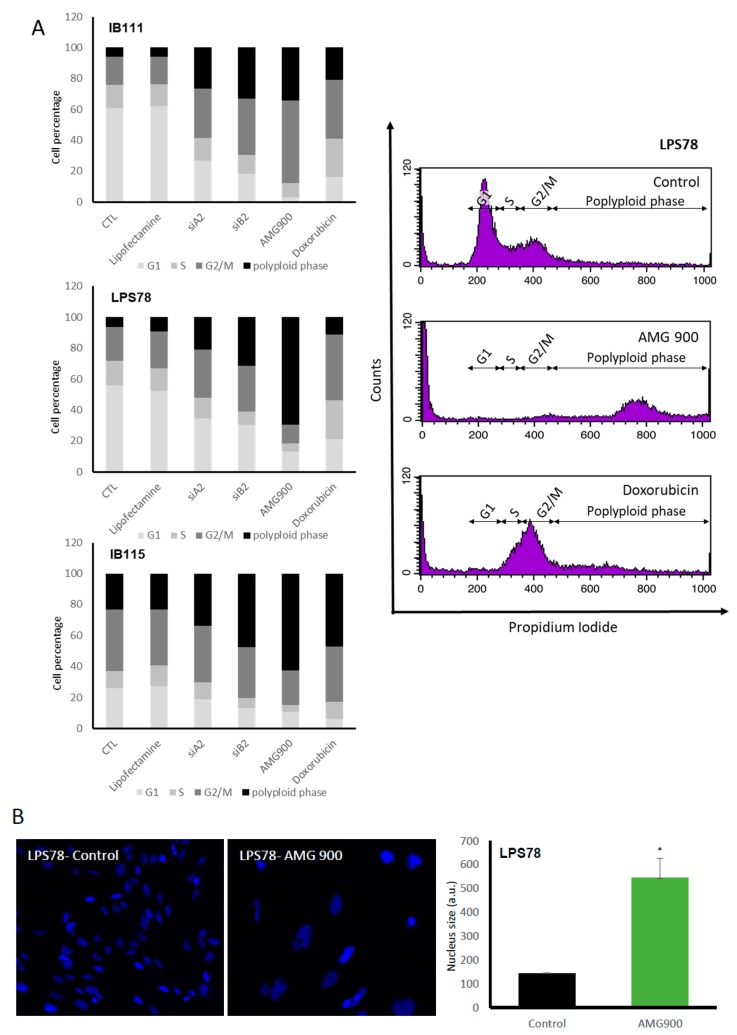
Modulation of AURKA and AURKB regulates cell cycle phases and leads to polyploidy. (**A**) Modification of cell cycle upon treatment in three LPS cell lines. Flow cytometry analyses are done on cells lines probed by PI, 72 h post-treatment: siRNA transfection, AMG 900 or doxorubicin. All the cell cycle phases for one condition are represented in vertical. (**B**) LPS78 nucleus aspect after 72H post-AMG 900 treatment. Nucleus were labelled with Hoechst 33342. Experiments were done in triplicate, with at least 100 events are analysed by experiment. Magnification ×400. (*: *p* < 0.05).

**Figure 7 cancers-12-00583-f007:**
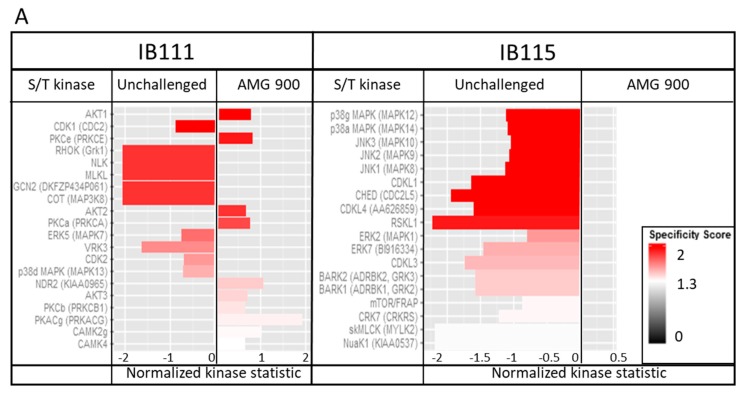
AMG 900 inhibits MAPK activity and modulates phosphorylation in sensitive cell line. (**A**) Variation of serine/threonine kinomic activity in resistant IB111 and sensitive IB115 with AMG 900 treatment. Cells were treated with IC_50_ of AMG 900 for one hour before protein extraction and analysis on PamChip. Kinases activities are compared between unchallenged and AMG 900-treated cells. Peptides phosphorylation leads to a serine/threonine kinases list presenting variation between both conditions. Graphic represent IB111 kinase activity in unchallenged (left) and AMG 900 treated-cell (right), intense red color correspond to high specificity score compare to white color. (**B**) Differential phosphorylation of MAPK in IB111 and IB115 cell lines, treated or not with AMG 900. Cells were treated with IC_50_ of AMG 900 for one hour before protein extraction and analysis by western-blot. Histograms are mean values of triplicate +/−SEM. (**: *p* < 0.01; ***: *p* < 0.001). More details of western blot, please view at the Appendix A.

**Figure 8 cancers-12-00583-f008:**
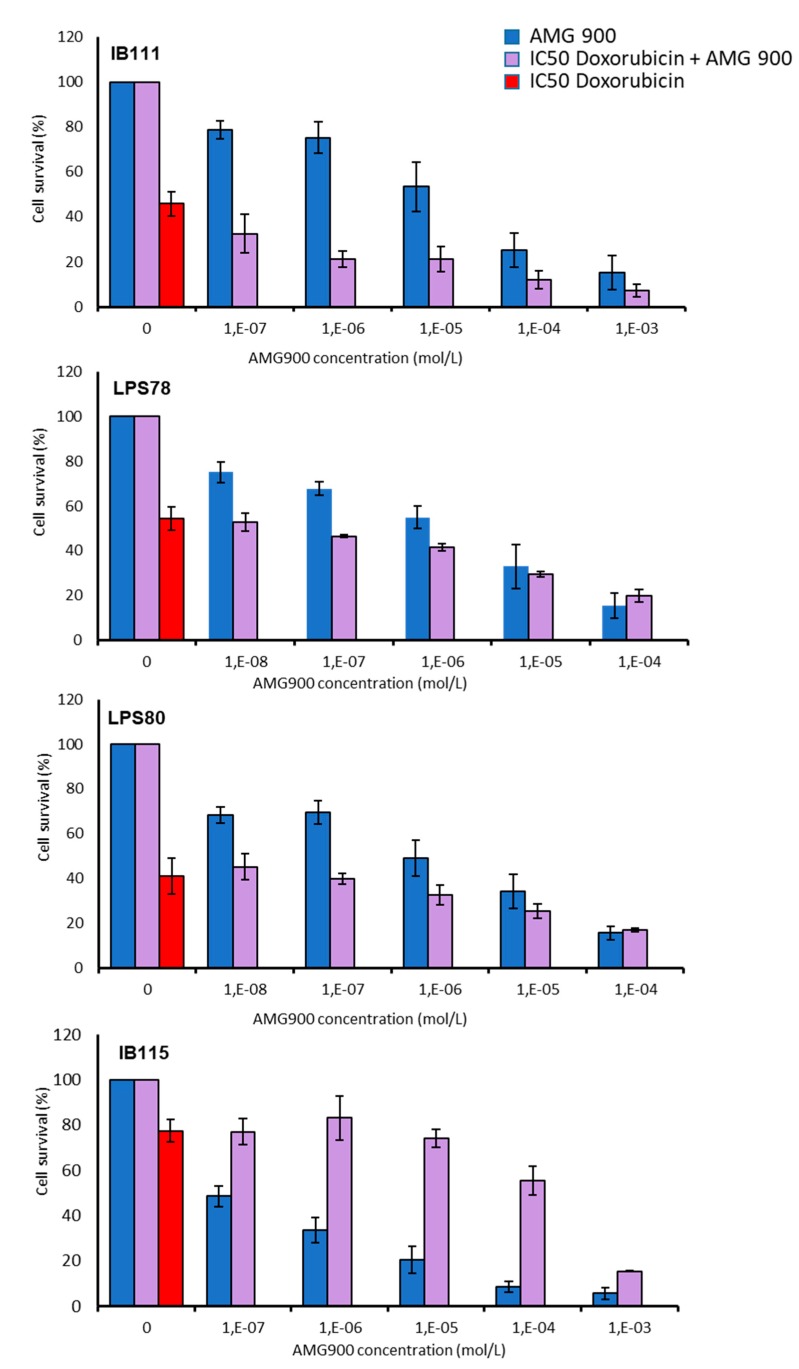
Doxorubicin-AMG 900 combined chemotherapy is efficient to induce cell mortality. LPS cells are challenged with IC_50_ of doxorubicin or a dose range of AMG 900 alone or in combination for 72H. Cell survival is assayed by MTT. Histograms are mean values of triplicate +/− SEM.

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
