# Peer review of "Pan Aurora Kinase Inhibitor: A Promising Targeted-Therapy in Dedifferentiated Liposarcomas With Differential Efficiency Depending on Sarcoma Molecular Profile"

_cancers, 2020, doi:10.3390/cancers12030583_

Round 1

Reviewer 1 Report

I´m sending my suggestions and comments in the pdf file.

Reviewer 2 Report

This paper highlight cytotoxic effect of AMG 900 on several STS, especially on dedifferentiated LPS, with an impact on clonogenicity, proliferation and DNA accumulation I think this is academically valuable research, but it needs to be revised in typographic format   L99-102 99-102 has two URLs but the page content is the same ABC symbols on the left side of Figure 1 suggest formal symbols, such as Figure 1 (A) and Figure 1 (b) The formal ABC symbol on the left side of Figure 2 is recommended, such as Figure 2 (A) and Figure 2 (b) Figure 4 has three pictures but is not labeled individually The ABC symbol on the left of Figure 5 is recommended to use formal symbols, such as Figure 5 (A) and Figure 5 (b) Figure 6A The resolution is too poor and needs to be corrected Figure 7A The resolution is too poor and needs to be corrected In line 450, it is recommended to add a graph of the relationship between the wavelength and the dissolution power, explaining the reason for choosing 530nm.

Reviewer 3 Report

The study by Matei et al with title “Pan Aurora kinase inhibitor: a promising targeted-therapy in dedifferentiated liposarcomas with differential efficiency depending on sarcoma molecular profile” is considering the use of AMG 900, an Aurora kinase inhibitor in the treatment of liposarcomas. The study is well designed and well written.  

Major revisions

The results from Figure 1 and Additional File 1 show differences in the significance of each kinase in clinical outcomes. Specifically, clinical data presented by the authors show a strong correlation between the expression of Aurora A and metastasis-free survival, while TCGA data show that Aurora B is the one significantly associated with overall and disease-free survival. This should be noted in the results section (the results are not similar as the authors mention). The authors should also discuss why there may be biological differences that lead to such differences. The cytometric analysis of Figures 5B and 6A cannot discriminate between apoptotic and necrotic cells. The experiments should be repeated by the more accurate Propidium iodide/annexin V analysis.

Minor revisions:

A final paragraph with the overall conclusions drawn from the study should be added, in order to summarize the interesting results. The discussion section is not easy to follow, since it lacks references to the results discussed. The figures containing the respective results should be mentioned. Editing: The authors should consider using the same typeface font throughout the text. In many cases (such as in some address lines, links etc.) a different font is used. The authors should also consider to crop the edges of most figures, since a grey frame is visible in most of them.

Round 2

Reviewer 3 Report

The manuscript has been substantially improved. I recommend acceptance after this round of revision.